**Open Peer Review** | *Clinical Microbiology* | Methods and Protocols

# A metagenomics method for the quantitative detection of bacterial pathogens causing hospital-associated and ventilator-associated pneumonia

S. Hauser,[1] V. Lazarevic,[2] M. Tournoud,[1] E. Ruppé,[2] E. Santiago Allexant,[3] G. Guigon,[3] S. Schicklin,[3] V. Lanet,[3] M. Girard,[2] C. Mirande,[4] G. Gervasi,[3] J. Schrenzel[2]

**ABSTRACT**   The management of ventilator-associated and hospital-acquired pneumonia requires rapid and accurate quantitative detection of the infecting pathogen(s). To achieve this, we propose a metagenomics next-generation sequencing (mNGS) assay that includes the use of an internal sample processing control (SPC) for the quantitative detection of 20 relevant bacterial species of interest (SOI) from bronchoalveolar lavage (BAL) samples. To avoid very major errors in the identification of respiratory pathogens due to "false-negative" cases, each sample was spiked with *Bacillus subtilis*, at a precisely defined concentration, using rehydrated BioBall. This SPC ensured the detection and quantification of the pathogen(s) at defined minimum concentrations. In the presented mNGS workflow, absolute quantification of *Staphylococcus aureus* was as accurate as quantitative PCR. We defined a metagenomics threshold at $5.3 \times 10^3$ genome equivalent unit per milliliter of the sample for each SOI, to distinguish colonization from higher amounts of pathogens that may be associated with infection. Complete mNGS process and metrics were assessed on 40 clinical samples, showing >99.9% sensitivity compared to microbial culture. However, 19 out of the 29 (66%) SOI detections above the metagenomics threshold were not associated with bacterial growth above classical culture-based clinical thresholds. Taxonomic classification of 7 (37%) of the "false-positive" detections was confirmed by finding specific 16S/MetaPhlAn2 markers, the 12 other "false-positive" detections did not yield enough reads to check their taxonomic classification. Our SPC design and analytical workflow allowed efficient detection and absolute quantification of pathogens from BAL samples, even when the bacterial DNA quantity was largely below the manufacturer's recommendations for NGS. The frequent "false-positive" detections suggested the presence of nonculturable cells within the tested BAL samples. Furthermore, mNGS detected mixed infections, including bacterial species not reported by routine cultures.

**IMPORTANCE**   The management of ventilator-associated pneumonia and hospital-acquired pneumonia requires rapid and accurate quantitative detection of the infecting pathogen. To this end, we propose a metagenomic sequencing assay that includes the use of an internal sample processing control for the quantitative detection of 20 relevant bacterial species from bronchoalveolar lavage samples.

**KEYWORDS**   clinical metagenomics, diagnosis, standardization, sample processing control, spike-in bacteria, calibrator, absolute quantification, metagenomics threshold, HAP-VAP

Address correspondence to S. Hauser, sebastien.hauser@biomerieux.com.

Some of the authors are employees of bioMérieux, a company creating and developing infectious disease diagnostics. No other potential conflicts of interest relevant to this article are reported.

Hospital-acquired pneumonia (HAP) and ventilator-associated pneumonia (VAP) are common nosocomial infections, predominantly of bacterial origin, causing high

morbidity rates (1). Optimal care for HAP/VAP requires prompt detection and identification of the causative pathogen.

Today, the "gold standard" method for the microbiological diagnosis of HAP/VAP is culture-based (2) and requires at least 24 h to identify the causative pathogen (3). Molecular methods that are independent of cultivability may improve speed and analytical sensitivity (4) over conventional tests (5–7). Clinical applications of metagenomic next-generation sequencing (mNGS) are currently drawing the attention of infectious diseases specialists (8, 9) and guidelines for NGS-based diagnostics have emerged (10–12). However, the technical requirements for NGS diagnostics remain to be defined. To control for the integrity of the reagents, equipment functionality, and the potential presence of inhibitors, the US Food and Drug Administration (FDA) (12) recommends the use of an internal control (IC), typically a "foreign sequence" co-extracted and co-analyzed with the sequences of the sample. Spiked nucleic acids (13) or exogenous bacteria (14) can be used as IC. Moreover, metagenomics analyses require control of the quantity and quality of the DNA extracted from the sample as well as the concentration, purity, and size distribution of the sequencing library (15). These control steps waste sample volume and are time-consuming, expensive, and tedious to make.

We propose the utilization of a sample processing control (SPC) that is spiked into the clinical sample and follows the entire analytical process. A positive SPC detection indicates that all steps of the sample processing were successful, and that sequencing data can be safely interpreted. This concept is widely used for integrated PCR cartridge systems such as GeneXpert (Cepheid), which contains *Bacillus globigii* spores or FilmArray (bioMérieux), which includes *Schizosaccharomyces pombe* cells. However, these SPC cannot be directly transposed from PCR to mNGS assays. As mNGS relies on random sequencing of DNA fragments, the detection of SPC competes with the detection of pathogens, commensal microbiota (flora), and the DNA of the patient. Therefore, the efficacy of the detection of SPC and pathogens depends also on the biological composition of each sample (relative abundance of microorganisms and the ratio of microbial to human cells) (16–18). In addition, clinical samples may contain large quantities of dead bacteria and extracellular bacterial DNA (19). The depletion of DNA from human cells and extracellular (human and bacterial) DNA is possible using selective lysis of host cells followed by endonuclease digestion. The SPC detection can then establish whether the mNGS is able to detect pathogens at a defined minimal concentration.

Clinically defined thresholds of pathogen concentrations are used to distinguish infection from asymptomatic colonization (20). For HAP/VAP diagnosis, culture-based thresholds are currently defined at $10^3$ colony-forming units (CFU)/ mL for mini-BAL and $10^4$ CFU/mL for BAL samples (21–23). Therefore, mNGS should also provide absolute quantification of the detected pathogens. Various experimental approaches have been proposed including cell counting by flow cytometry (24), normalization of bacterial relative abundance based on defined cell numbers that are spiked into the samples before nucleic acid isolation (25), or use of spiked nucleic acids (26). However, these designs cannot provide an absolute quantification of pathogens.

To control all processing steps and to quantify the abundance of the detected pathogen(s), we spiked samples with rehydrated BioBall (bioMérieux) as an SPC. Diagnostic capabilities of the mNGS workflow were assessed in bronchoalveolar lavage (BAL) samples using a panel of bacterial species that commonly cause HAP/VAP infection.

## MATERIALS AND METHODS

### Selection of 20 bacterial pathogens for HAP/VAP diagnosis

We defined a panel of bacterial pathogens frequently involved in HAP/VAP in immunocompetent patients. This panel, composed of 20 bacterial species of interest (SOI), is listed in Table 1.

**TABLE 1** List of the 20 pathogens composing the HAP/VAP panel

| Enterobacterales | Gram-negative bacilli | Gram-positive cocci |
|---|---|---|
| *Escherichia coli* | *Pseudomonas aeruginosa* | *Staphylococcus aureus* |
| *klebsiella oxytoca* | *Stenotrophomonas maltophilia* | *Streptococcus pneumoniae* |
| *klebsiella pneumoniae* | *Acinetobacter baumannii* | |
| *klebsiella aerogenes* | *Legionella pneumophila* | |
| *Enterobacter cloacae* | *Haemophilus influenzae* | |
| *Serratia marcescens* | | |
| *Proteus mirabilis* | | |
| *Proteus vulgaris* | | |
| *Hafnia alvei* | | |
| *Citrobacter freundii* | | |
| *Citrobacter koseri* | | |
| *Morganella morganii* | | |
| *Providencia stuartii* | | |

## Collection of fresh BAL and mini-BAL samples

BAL or mini-BAL fluids were obtained at Geneva University Hospitals (HUG, Switzerland) from patients suspected of pneumonia, with no other specific clinical or demographic inclusion criteria. The samples were sent to the bacteriology laboratory for routine culture analysis on solid media (blood agar, chocolate agar, Mac Conkey agar, and Columbia colistin-nalidixic acid agar). The 20 SOI may grow on these media used for culturing the respiratory specimens except for *Legionella pneumophila*. Exceptionally, *L. pneumophila* can be grown on chocolate agar, albeit with extended incubation time, and possibly when using a nutrient-rich specimen (e.g., a biopsy) (27). In our study, solid media were incubated for 48 h; hence, *L. pneumophila* would not be isolated, even if nutrients were putatively brought by the specimen. Following plating and a 48-h incubation at 35°C in a 5% $CO_2$ enriched atmosphere, CFU were counted.

Strain identification of the predominant colony morphotypes was performed using matrix-assisted laser desorption ionization-time of flight mass spectrometry (MALDI Biotyper, Bruker Daltonics, Bremen, Germany) according to the manufacturer's instructions. For each sample, a list of the SOI detected is provided along with an estimate of their concentration rounded to the nearest log10 value.

Two sets of samples were collected, the first set named "training set" included 45 samples (22 BAL and 23 mini-BAL), and the second set, named "validation set," included 40 samples (33 BAL and 7 mini-BAL).

## Sample processing control

To control the whole mNGS process, we spiked the samples with spores of the Gram-positive bacterium *Bacillus subtilis* as an SPC. The SPC sequences are detected by our pipeline with an accuracy of 99.95%, without cross-identification of the SOI (see Fig. S1).

The *B. subtilis* strain ATCC 19659 BioBall MultiShot 10E8 (bioMérieux, Catalog #416721) was re-hydrated in 1.1 mL of phosphate buffered saline (PBS) to provide a solution of $10^8$ CFU/mL ± 20%. After a 100-fold dilution in PBS, 10 µL of the suspension was added to 600 µL of the (mini-)BAL fluid. The final SPC concentration within a sample was $1.7 \times 10^4$ CFU/mL, close to the culture-based diagnostic threshold ($10^4$ CFU/mL) for HAP/VAP.

## Sample preparation and DNA extraction

Six hundred microliters of BAL supplemented with SPC was mixed with 6 mL of Tris-HCl 50 mM (pH 8) containing polyethylene glycol (PEG) 4% and saponin 0.08% and incubated at room temperature for 10 min for the selective lysis of human cells. After saponin treatment, the non-lysed cells were pelleted (12,000 *g* for 10 min) and treated with DNase I (5,000 U, 37°C, for 15 min). The DNase was inactivated by heating (80°C for 10 min)

and the addition of EDTA (10 mM final concentration). The sample was then added to a tube containing a mix of 1 mm glass beads (600 mg per 1.5 mL tube) and 0.1 mm zirconia/silica beads (150 mg per 1.5 mL tube) and bacteria were disrupted by shaking for 20 min on a vortex with a horizontal tube holder. Nucleic acids were extracted from the lysate on an easyMAG platform (bioMérieux) using the generic protocol (V2.0.1). Elution was carried out in a volume of 25 µL, and the extracts were stored at –20°C. Human and bacterial DNA load were determined as described previously (18).

## Library preparation and MiSeq sequencing

Libraries were prepared for 2 × 250 paired-end sequencing with a modified protocol of the Nextera XT DNA Library Preparation Kit (Illumina).

To allow efficient preparation of DNA libraries for all extracts, the protocol was adapted to run with only a few picograms of DNA (28, 29). The following steps were added to the standard protocol: (i) when the concentrations of extracted DNA were above 0.2 ng/µL, 1 ng was used for the preparation of the library; otherwise, a maximum volume of 5 µL of DNA extract was used, regardless of its concentration; (ii) after tagmentation, 16 amplification cycles were applied to the DNA library in order to obtain enough material for sequencing; and (iii) the library was purified using 25 µL of AMPure XP beads (Beckman Coulter). Indexed libraries prepared from two samples were pooled at equal quantities before sequencing on the MiSeq platform (see Fig. 1) with the MiSeq Reagent kit V3 (Illumina) following the manufacturer's instructions.

## Bioinformatics pipeline

For the identification of pathogens, sequence reads were analyzed using the metagenomics pipeline described by Jaillard et al. (30) and Tournoud et al. [awaiting peer review (31)]. The detection of antibiotic resistance genes was not assessed in this study.

Briefly, the bioinformatics pipeline consisted of the following steps: quality control of the reads, trimming and filtering of poor quality reads, and taxonomic read binning with Kraken (32) using an internal reference database including more than 10,000 genomes from the 20 SOIs, *B. subtilis*, bacteria found in the lung and oral cavity, and the human genome (*Homo sapiens* genome assembly GRCh37). Bacterial genomes are both public [e.g., PATRIC (33), RefSeq (34), FDA ARGOS (35)] and private (strains sequenced from BAL samples). The median number of genomes per species was 42 with inter-quartile-range equal to (19–297), and the total number of genomes belonging to the flora was 7,593. The performance of taxonomic classification of simulated sequence reads of SOI, *B. subtilis,* and Human from the internal reference database is reported in Fig. S1.

Pathogen detection and quantification relied on the number of reads attributed to each SOI and the SPC. To avoid spurious pathogen detection due to erroneous taxonomic classification, reads associated with an SOI with an average genome sequencing depth >1× were assembled. Assembly was performed using the idba_ud500 assembler (IDBA-UD 1.1.1) (36), with the following parameters: —mink 40 —maxk 250 —min_pairs 2. To confirm species identification, the assemblies were aligned with BLAST (37) against a pathogen marker database built by selecting clade-specific MetaPhlAn2 (38) and 16S markers (for *Hafnia alvei*, *Proteus vulgaris*, and *Morganella morganii*, for which no MetaPhlAn2 markers were available) to allow unambiguous taxonomic assignment to the 20 SOIs. The BLAST algorithm was run with 75% coverage and 97% identity. When at least one marker of the tested SOI was detected, the taxonomic assignment of the sequences to the SOI was confirmed. When a marker from another species was detected and not the one of the tested SOI, the tested SOI was invalidated to avoid false-positive detection induced by misclassification of the reads. When no pathogen marker was detected, mainly because of low-coverage assemblies, no confirmation or invalidation of taxonomic classification to SOI was reported.

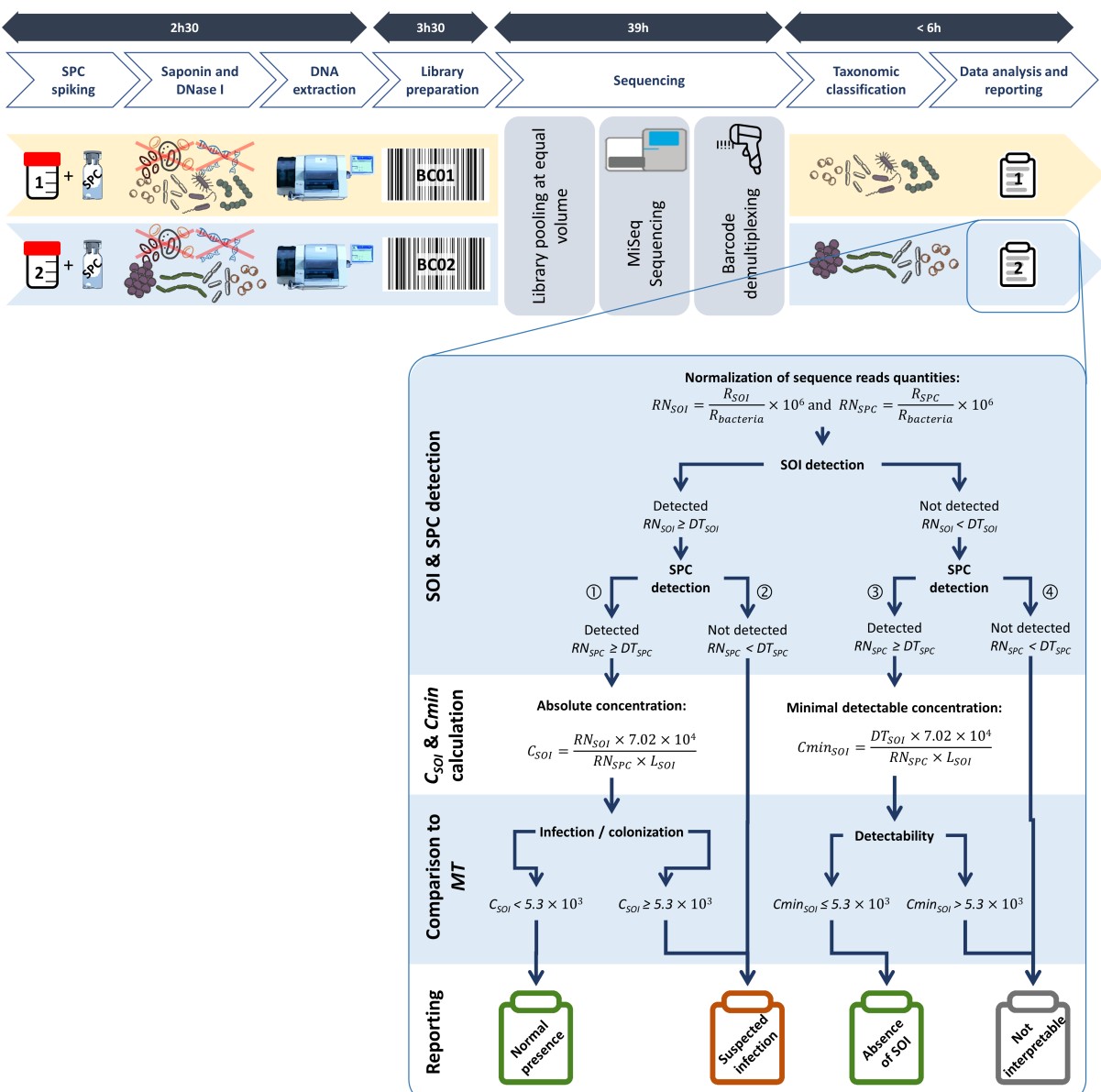

**FIG 1** Complete workflow for quantitative detection of pathogen causing HAP/VAP by metagenomics. The duration of the key steps is shown at the top of the figure. $RN_{SOI}$ and $RN_{SPC}$ are normalized quantities in reads per million of bacterial reads classified as SOI and SPC, respectively. $DT_{SOI}$ and $DT_{SPC}$ are thresholds of detection of the normalized quantity of reads associated with SOI and SPC, respectively. $L_{SOI}$ is the genome size of SOI (in Mb). $C_{SOI}$ is the calculated absolute concentration of SOI in a sample in genome equivalent per milliliter (GEq/mL). $Cmin_{SOI}$ is the minimal detectable concentration of SOI in a sample (in GEq/mL) for a given mNGS run.

## Sequence data analysis and interpretation

The complete mNGS workflow is presented in Fig. 1. For each sample, all SOIs were analyzed separately as independent detection assays. Detailed procedures and calculations are provided in the supplemental material. Briefly, data interpretation consists of four steps.

### Step 1: SOI and SPC detection

The numbers of reads classified to a bacterial species of interest ($R_{SOI}$) or to the SPC ($R_{SPC}$) are normalized to the number of millions of reads associated with all bacterial

species in our database ($R_{bacteria}$). The normalized values ($RN_{SOI}$ and $RN_{SPC}$) are reported in reads per million of bacterial reads (RPMB).

We defined detection thresholds (DTs) to differentiate background from "true" taxonomic classification of sequence reads as SPC or SOI within sequenced sample (see supplemental material). These DTs represent the minimal quantity of classified sequence reads normalized per million of bacterial reads required to report SPC or SOI as detected.

### Step 2: Calculation of absolute concentration or minimal detectable concentration of SOI

As SPC is added at a defined concentration ($C_{SPC} = 1.7 \times 10^4$ CFU/mL) within each sample, it can be used as a calibrator.

- When SOI and SPC are detected (Fig. 1), the absolute concentration of SOI ($C_{SOI}$) can be calculated as

$$C_{SOI} = \frac{RN_{SOI} \times 7.02 \times 10^4}{RN_{SPC} \times L_{SOI}}$$

- When a SOI is not detected (Fig. 1), its minimal detectable concentration ($Cmin_{SOI}$) can be calculated if SPC has been detected

$$Cmin_{SOI} = \frac{DT_{SOI} \times 7.02 \times 10^4}{RN_{SPC} \times L_{SOI}}$$

$RN_{SOI}$ and $RN_{SPC}$ correspond to the normalized quantity of sequence reads associated to SOI and SPC, respectively. $L_{SOI}$ corresponds to the genome size of SOI in megabases. $7.02 \times 10^4$ is the product of SPC spiked concentration ($1.07 \times 10^4$ GEq/mL) and its genome size in megabases (4.13 Mb).

### Step 3: Comparison to the metagenomics threshold

A metagenomics threshold (MT) was defined (see supplementary material) as the concentration in genome equivalent (GEq) of SOI above which infection can be suspected. When a SOI is not detected (Fig. 1), the comparison of MT to the calculated $Cmin_{SOI}$ makes it possible to differentiate the absence of SOI in the sample from the inability to detect this SOI at a concentration defined by the MT.

### Step 4: Reporting

In each sample, the result of the detection of each SOI is reported separately. The rules of interpretation (see Fig. 1) define four possible reporting outcomes:

- Suspected infection, when an SOI is detected and quantified at or above MT or when the SOI is detected, but the SPC is not.

- Suspected colonization, when an SOI is detected and quantified below the MT.

- Absence of detection, when the SOI is not detected and the calculated $Cmin_{SOI}$ is <MT.

- Not interpretable, when both SOI and SPC are not detected or when the calculated $Cmin_{SOI}$ is >MT.

### PCR quantification of *Staphylococcus aureus*

Quantitative PCR targeting the *spa* gene was used to quantify *S. aureus*. Master mixes were prepared by mixing 0.2 µM of forward primer 1914F (CAGCAAACCA TGCAGATGCT AA), 0.2 µM of reverse primer 1992R (ACAGTTGTAC CGATGAATGG ATTTT), and 0.1 µM of

probe 1945T (AGCATTACCA GAAACT) (39) in the ABsolute QPCR Mix, no ROX (Thermo Scientific). One microliter of DNA extract was added in a final reaction volume of 50 µL. The amplification and the real-time reading of the fluorescent signal were carried out on the CFX96 Touch Real-Time PCR Detection System (Bio-Rad) using the following program: 95°C for 15 min; 42 cycles (95°C, 15 s; 60°C, 1 min).

A standard curve was produced from the genomic DNA of *S. aureus* strain MW2 at concentrations ranging between 1 pg and 1 ng and corresponding to $3.1 \times 10^2$ to $3.1 \times 10^5$ GEq.

## RESULTS

### Absolute quantification of *S. aureus*

We compared the quantity of *S. aureus* obtained by our mNGS approach to the results of qPCR and culture quantifications (Table 2) on 11 samples from the "training set." These samples were either culture-positive for *S. aureus* ($n = 10$) or culture-negative but with *S. aureus* detection by mNGS (sample T07).

The comparison of the nearest log10 of concentrations determined by mNGS (in GEq/mL) and microbial culture (in CFU/mL) (Fig. 2A) showed moderate correlation (slope = 0.6087) with a weak linear association ($R^2 = 0.3043$), which was statistically insignificant ($P = 0.156$). The comparison of the log10 values of concentrations determined by mNGS with those obtained from quantitative PCR (Fig. 2B) revealed a strong correlation (slope $\approx$ 1), strong linear association ($R^2 = 0.8392$), and statistical significance ($P < 0.01$).

Moreover, for sample T07, which was culture-negative, both qPCR and mNGS detected and quantified *S. aureus* above $10^4$ GEq/mL.

**TABLE 2** Comparison of *S. aureus* quantification by microbial culture, qPCR, and mNGS

| ID | Sample type | Bacterial species | Quantification (log10)[a] | | | 16S MetaPhlAn2 markers |
|----|-------------|-------------------|-------------------|------|----------------|------------------------|
| | | | Microbial culture | mNGS | qPCR (*S. aureus*) | |
| T01 | Mini-BAL | *S. aureus* | >5 | 7.4 | 7.2 | + |
| T03 | Mini-BAL | *S. aureus* | 3 | 6.2 | 6.3 | + |
| | | *S. pneumoniae* | 3 | 4.9 | | |
| T07 | Mini-BAL | *S. aureus* | NEG | 4.0 | 5.0 | + |
| T17 | BAL | *K. pneumoniae* | NEG | 7.5 | | + |
| | | *S. aureus* | 2 | 5.1 | 4.3 | + |
| T18 | BAL | *E. coli* | >5 | 7.0 | | + |
| | | *S. aureus* | >5 | 6.3 | 6.3 | + |
| T19 | BAL | *S. aureus* | 5 | 6.3 | 5.5 | + |
| T26 | BAL | *E. coli* | >5 | 7.7 | | + |
| | | *P. mirabilis* | >5 | 3.9 | | |
| | | *S. aureus* | >5 | 5.6 | 5.0 | + |
| T28 | BAL | *H. influenzae* | >5 | 7.1 | | + |
| | | *P. vulgaris* | 4 | 4.0 | | |
| | | *S. aureus* | 5 | 5.0 | 4.5 | + |
| T31 | BAL | *H. influenzae* | | 3.8 | | + |
| | | *S. aureus* | 5 | 6.0 | 5.2 | + |
| | | *S. maltophilia* | 5 | 7.9 | | + |
| T43 | BAL | *H. influenzae* | | 5.7 | | + |
| | | *P. mirabilis* | >5 | 7.7 | | + |
| | | *S. aureus* | 2 | 2.4 | 3.0 | |
| T44 | BAL | *S. aureus* | 4 | 6.1 | 5.9 | + |
| T45 | BAL | *S. aureus* | 4 | 3.7 | 3.6 | + |

[a]Log10 of quantified pathogen concentrations in CFU/mL (for microbial culture) and GEq/mL (for mNGS and qPCR) is presented in the corresponding columns.

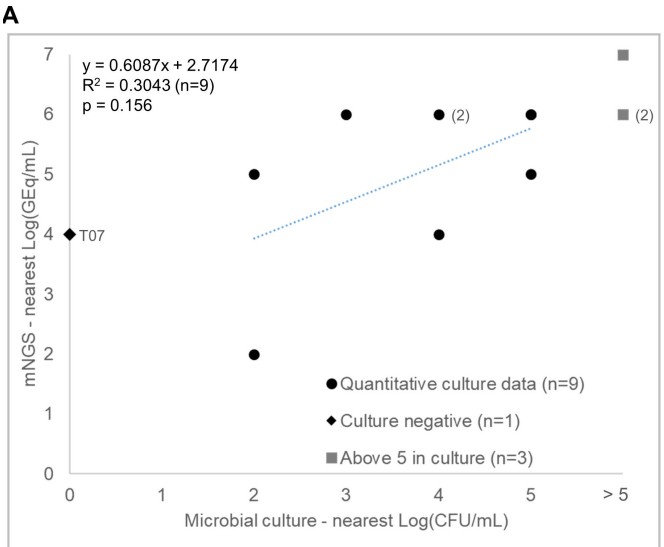 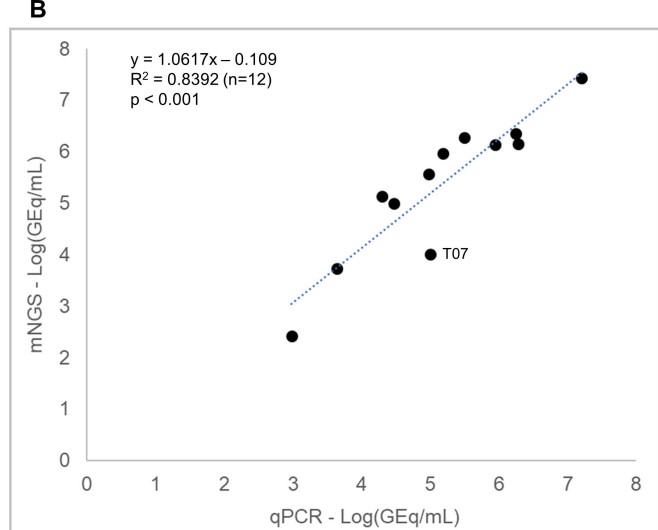

**FIG 2** Comparison of *S. aureus* quantification. (A) Comparison of quantification in GEq/mL obtained through mNGS to CFU/mL determined by microbial culture. (B) Comparison of quantification by mNGS to quantitative PCR, both in GEq/mL.

## Validation of quantitative detection of SOI

We evaluated the complete mNGS workflow (Fig. 1) with a "validation set" of 40 samples, comparing the quantitative detection of SOIs by culture and mNGS (Table 3). Twenty samples (50%) tested negative for all SOIs by culture. Bacterial culture and mNGS identified other species outside the study panel. However, as we did not define the detection threshold for these species and their MetaPhlAn markers were not considered, our data analysis focused on the 20 SOIs.

### *Control of the sample processing*

The SPC detection failed ($RN_{SPC} < DT_{SPC}$) in four samples (10%) in which at least one SOI was detected as a probable infecting agent (samples 3, 7, 23, and 35 in Table 3). In these samples, non-detected SOIs are reported as "not interpretable" (Fig. 1), as the minimal detectable concentration may be above the MT.

The SPC detection ($RN_{SPC} \geq DT_{SPC}$) in 36 samples (90%) provided conditions for mNGS results' validation. The presented mNGS workflow was able to process culture-negative samples, e.g., sample 36 (Table 3) that contained as low as 0.5 pg/µL DNA in the extract and generated only 4,850 reads. The detection of SPC and estimated $Cmin_{SOI}$ below MT allowed for validating the sequencing results and reporting a negative detection for these samples. However, we were not able to validate 10.8% (75/697) of negative mNGS SOI detections because the calculated $Cmin_{SOI}$ exceeded the MT (Fig. 1). It mainly concerned *E. coli*, *K. pneumoniae*, *P. vulgaris*, *S. aureus*, and *S. pneumoniae* (Table 3), which had high $DT_{SOI}$ values (see Fig. S4).

### *Quantitative detection of the SOI panel*

mNGS results were compared to the current "gold standard" for HAP/VAP diagnosis, i.e., the microbial culture (Fig. 3). Ten "true positive" detections of probable infection by SOIs and no "false-negative" results (Table 3; Fig. 3) revealed a test sensitivity above 99.9%. With 19 "false-positives" (Table 3; Fig. 3), the test specificity was 96.6% but the "false discovery rate" reached 65.5%. Importantly, we confirmed proper taxonomic classification of 26.3% (5/19) of the "false-positive" detections by finding specific 16S/MetaPhlAn2 markers. The 14 other "false-positive" detections did not yield enough reads to check their taxonomic classification by 16S/MetaPhlAn2 marker search.

**TABLE 3** Detection of SOI(s) above clinical threshold by microbial culture or above MT by mNGS in the "validation set"[a]

| N° | Sample type | Bacterial species | Culture | mNGS | 16S/MetaPhlAn2 markers | SPC detected | Comments |
|---|---|---|---|---|---|---|---|
| 1 | BAL | *S. pneumoniae* | > 5 | 4.8 | + | + | |
| 2 | Mini-BAL | | | | | + | |
| 3 | BAL | *P. aeruginosa* | > 5 | >MT | + | − | Negative detections of other SOI cannot be validated. |
| 4 | BAL | | | | | + | Cmin > MT for *E. aerogenes, E. coli, H. influenzae, K. pneumoniae, P. vulgaris, S. aureus,* and S. *pneumoniae*. |
| 5 | BAL | | | | | + | |
| 6 | BAL | *P. aeruginosa* | 2 | 4.7 | | + | Cmin > MT for *E. aerogenes, E. coli, H. influenzae, K. pneumoniae, P. vulgaris, S. aureus,* and S. *pneumoniae*. |
| | | *S. marcescens* | 2 | 5.5 | + | | |
| 7 | Mini-BAL | *S. marcescens* | 2 | >MT | + | − | Negative detections of other SOI cannot be validated. |
| 8 | Mini-BAL | | | | | + | Cmin > MT for *A. baumannii, C. koseri, E. aerogenes, E. coli, H. influenzae, K. oxytoca, K. pneumoniae, P. vulgaris, S. aureus, S. maltophilia,* and *S. pneumoniae*. |
| 9 | BAL | *S. pneumoniae* | 5 | 6.0 | + | + | Cmin > MT for *E. coli* and *P. vulgaris*. |
| 10 | BAL | | | | | + | |
| 11 | BAL | *L. pneumophila* | + | 1.4 | | + | |
| 12 | BAL | | | | | + | |
| 13 | BAL | *S. aureus* | 5 | 5.9 | + | + | Cmin > MT for *P. vulgaris*. |
| 14 | BAL | | | | | + | |
| 15 | BAL | | | | | + | |
| 16 | Mini-BAL | | | | | + | |
| 17 | BAL | | | | | + | |
| 18 | BAL | | | | | + | |
| 19 | BAL | *S. pneumoniae* | | 4.1 | | + | Cmin > MT for *E. coli* and *P. vulgaris*. |
| 20 | BAL | *P. aeruginosa* | 5 | 5.6 | + | + | Cmin > MT for *P. vulgaris*. |
| 21 | Mini-BAL | | | | | + | |
| 22 | BAL | *S. maltophilia* | 5 | 7.3 | + | + | Cmin > MT for *A. baumannii, C. koseri, E. aerogenes, E. coli, H. influenzae, K. oxytoca, K. pneumoniae, M. morganii, P. vulgaris, P. stuartii, P. aeruginosa, S. aureus,* and *S. pneumoniae*. |
| 23 | Mini-BAL | *P. aeruginosa* | > 5 | >MT | + | − | Negative detections of other SOI cannot be validated. |
| | | *S. aureus* | 2 | Invalid | | | |
| 24 | BAL | | | | | + | |
| 25 | BAL | | | | | + | |
| 26 | BAL | | | | | + | |
| 27 | BAL | *C. freundii* | | 4.5 | | + | Cmin > MT for *A. baumannii, C. koseri, E. aerogenes, E. coli, H. influenzae, M. morganii, P. vulgaris, P. stuartii, S. aureus,* and *S. pneumoniae*. |
| | | *H. alvei* | | 4.8 | | | |
| | | *K. oxytoca* | | 4.8 | | | |
| | | *K. pneumoniae* | | 6.1 | + | | |
| | | *P. aeruginosa* | | 4.5 | | | |
| | | *S. marcescens* | | 5.0 | | | |
| | | *S. maltophilia* | | 4.4 | | | |
| | | *E. cloacae* complex | > 5 | 7.1 | + | | |
| 28 | BAL | *S. marcescens* | 2 | 4.2 | + | + | |
| 29 | BAL | *E. aerogenes* | 3 | 6.0 | + | + | Cmin > MT for *E. coli, H. influenzae, K. pneumoniae, P. vulgaris,* and *S. pneumoniae*. |
| | | *S. aureus* | 2 | 5.5 | + | | |
| | | *K. oxytoca* | | 3.8 | | | |
| 30 | Mini-BAL | | | | | + | |
| 31 | BAL | *H. influenzae* | > 5 | 4.8 | + | + | |
| 32 | BAL | | | | | + | |
| 33 | BAL | | | | | + | |

*(Continued on next page)*

TABLE 3 Detection of SOI(s) above clinical threshold by microbial culture or above MT by mNGS in the "validation set"[a] (Continued)

| N° | Sample type | Bacterial species | Culture | mNGS | 16S/MetaPhlAn2 markers | SPC detected | Comments |
|---|---|---|---|---|---|---|---|
| 34 | BAL | S. marcescens | 3 | 4.8 | + | + | Cmin > MT for E. coli, K. pneumoniae, P. vulgaris, and S. pneumoniae. |
| 35 | BAL | P. aeruginosa | **5** | >MT | + | − | Negative detections of other SOI cannot be validated. |
| 36 | BAL | | | | | + | |
| 37 | BAL | | | | | + | Cmin > MT for E. coli and P. vulgaris are above MT. |
| 38 | BAL | S. maltophilia | 3 | 4.4 | | + | |
| 39 | BAL | P. aeruginosa | **> 5** | 7.0 | + | + | Cmin > MT for A. baumannii, C. koseri, E. aerogenes, E. coli, H. influenzae, K. oxytoca, K. pneumoniae, P. vulgaris, S. aureus, and S. pneumoniae. |
| | | S. maltophilia | | 4.2 | | | |
| 40 | BAL | P. aeruginosa | 2 | 3.8 | | + | |

[a]Log10 of quantified pathogen concentrations in CFU/mL (for microbial culture) and GEq/mL (for mNGS) is presented in the corresponding columns. >MT means that SOI was detected but not the SPC, suggesting that detected SOI were likely present at a concentration above MT (Fig. 1②). (+) in 16S/MetaPhlAn2 markers column means that taxonomic classification of sequence reads to a bacterial species is confirmed by 16S/MetaPhlAn2 markers search. Values in bold correspond to quantified pathogen concentrations exceeding the clinical threshold (in Culture column) or metagenomics threshold (in mNGS column).

Sample 27, which was culture positive for *E. cloacae* complex above $10^5$ CFU/mL, led to seven "false-positive" detections (Table 3) corresponding mostly to Enterobacterales. The presence of *K. pneumoniae*, detected at a concentration above $10^6$ CFU/mL, was confirmed by the 16S/MetaPhlAn2 markers search. But the other "false-positive" SOI detections were quantified at 10- to 100-fold lower concentrations relative to *K. pneumoniae* and had too few reads for 16S/MetaPhlAn2 confirmation.

### Detection of co-infections

The presence of a single infecting agent at a high concentration may preclude the detection of other SOIs and the SPC and may limit the detection of co-infections by mNGS. However, we were able to detect two (samples 6 and 39) or three (sample 29) co-infecting pathogens with absolute concentrations differing by up to three orders of magnitude (Table 3). None of these mixed infections were reported by microbial cultures.

## DISCUSSION

Here, we propose spiking BioBall into (mini-)BAL samples to control processing and provide absolute SOI quantification in the mNGS workflow. We selected *B. subtilis* as SPC because of its rare natural presence in BAL samples and the ability of mNGS to distinguish its sequence reads from those of SOIs, commensal flora, and the human genome (see Fig. S1). The quantitative metagenomics assay and metrics (Fig. 1) were developed using a "one system" approach. This means that all individual steps, from sample preparation to results reporting, are controlled by the SPC, thus eliminating the need for the fastidious steps required to control and quantify extracted DNA and sequencing libraries.

In a mNGS run, the detection limit of a SOI can be impacted by intrinsic factors including genome length and efficiency of its DNA extraction as well as extrinsic factors such as the accuracy of taxonomic classification and sample composition (host cell load, relative abundance of microorganisms, peculiarities of genome sequences for certain microorganisms) (19). To avoid reporting false-negative results, the detection of a SOI should only be reported as negative when Cmin ≤ MT (Fig. 1). This was especially useful for the analysis of culture-negative samples in which, after patient's DNA removal, remaining DNA quantities were significantly below those recommended for library preparation and sequencing. While modifications of the library preparation protocol allowed the sequencing of these samples, the number of reads remained very low. By detecting the SPC and calculating the Cmin for each SOI, we validated the ability of the test to detect an SOI at a concentration at least equal to the metagenomics threshold. Unfortunately, we were unable to validate 20.7% of negative SOI detections: in the

| | | mNGS detection | | | |
|---|---|---|---|---|---|
| | | Above or equal to $MT$ | No detection or below $MT$ | No detection with no SPC detection or $Cmin_{SOI}$ above $MT$ | |
| **Culture based detection** | Above or equal to clinical threshold | True Positive (TP)  10 | False negative (FN)  0 | Not interpretable  0 | Sensitivity $\dfrac{TP}{TP + FN}$  100.0 % |
| | Negative or below clinical threshold | False positive (FP)  19 | True negative (TN)  548 | Not interpretable  143 | Specificity $\dfrac{TN}{TN + FP}$  96.6 % |
| | | False discovery rate $\dfrac{FP}{FP + TP}$  65.5 % | False omission rate $\dfrac{FN}{FN + TN}$  0 % | | |

**FIG 3** Comparison of mNGS to culture detections of 18 SOIs from all of the 40 samples from validation set.

four samples where SPC were not detected, we reported a probable infection by the detected SOI, but the 72 negative detections could not be validated. In the samples where SPC were detected, 75 negative SOI detections had an estimated Cmin > MT. High Cmin values resulted mainly from $DT_{SOI}$ above 1,000 RPMB. To reduce $DT_{SOI}$, it may be necessary to decrease the level of false taxonomic classification of reads by improving the specificity of classification algorithms and by using a curated reference sequence database. The example of *E. cloacae* illustrates difficulties in interpreting results due to a high detection threshold (>1,000,000 RPMB) likely caused by a contaminated reference database and, as a consequence, a high rate of misclassifications.

It is interesting to note that despite the competition in the detection of SOI, our mNGS process allowed us to detect co-infections by two or three SOIs with concentrations ranging over three orders of magnitude. These co-infections were not detected by the routine cultures. Our results are consistent with previous observations that mNGS assays could be more effective in characterizing polymicrobial infections when compared to traditional culture-based methods (40).

Qualitative detection of microorganisms by mNGS can reflect resident microbiota, transient colonization, sample contamination, and/or infection. To differentiate the asymptomatic presence of bacteria from probable infection, absolute quantification has to be performed and compared to defined clinical decision thresholds (5, 41). For that purpose, we used the counts of reads assigned to SPC as a calibrator for the

quantification of SOIs (Fig. 2). Using *S. aureus* as an example, the results of absolute quantification by mNGS (Fig. 2B) were comparable to those of qPCR (42).

Clinical microbiology laboratories have defined clinical decision thresholds for the HAP/VAP causative pathogen(s) in CFU/mL (mini-BAL: $10^3$ CFU/mL and BAL: $10^4$ CFU/mL) (20). We did not find an obvious correlation between the number of genomes quantified by mNGS or qPCR and CFU counts from culture plates (Fig. 2A). This may have different causes such as a lack of precision of the culture report that provides the concentration of CFU at the nearest log10 level. However, even when reducing the precision of mNGS quantification by rounding to the nearest log10 value, the correlation between microbial culture and mNGS remains moderate and statistically insignificant (Fig. 2A). Other contributing factors such as the presence of viable but non-culturable (VBNC) (43, 44) cells in the sample, or the clustering of bacterial cells, can lead to an underestimation of the culture results relative to mNGS. Therefore, we defined the metagenomics threshold at $5.3E \times 10^3$ GEq/mL to differentiate, similar to the clinical decision thresholds, the asymptomatic presence of bacteria from infection (5, 41, 45–47).

Assessment of our mNGS process and defined metrics on the HAP/VAP panel showed good diagnostic capabilities (specificity: 96.6%; sensitivity: >99.9%), albeit with a "false discovery rate" of 65.5%. To avoid "false-positive" detections that may result from a lack of accuracy in the taxonomic classification, additional control methods should be considered. In this study, we confirmed correct sequence classification to the SOI for at least 26% of "false-positive" detections using 16S/MetaPhlAn2 markers detection by BLAST. For the other "false-positive" results, the number of reads was insufficient for the sequence assembly required for the BLAST-based 16S/MetaPhlAn2 markers search. Therefore, other controls should be considered such as the removal of reads that are stacked on a single location and share identity with the human genome or with commensal flora, as suggested by Uprety et al. (48). Nevertheless, in the presented mNGS workflow, no "false-positive" tests seemed to result from (k-mer based) misclassification of sequence reads, as they were not invalidated by finding 16S/MetaPhlAn2 markers from a different species. As the DNase I treatment of samples before the bacterial lysis step may remove extracellular DNA and genomic DNA from dead bacteria (6, 49), the new detections might reflect the presence of VBNC (6, 7) or antibiotic persister bacterial cells (44) within BAL samples. Currently, we have defined an MT that we apply to all bacterial species of our HAP/VAP panel, by analogy with cultures where a single clinical decision threshold is applied to all bacterial pathogens. It was suggested by Jahn et al. (41) that specific MT should be established for each bacterium depending on its pathogenicity. The presence of VBNC and antibiotic persisters could also be taken into account for the setting of specific MT. However, the evaluation of specific thresholds for each SOI would require large numbers of samples and clinical data, which were not available to us at the time of this study.

## Conclusions and perspectives

We present a new clinical metagenomics workflow for the detection of the causative pathogen(s) of HAP/VAP (Fig. 1). It includes the use of an SPC to ensure that all steps, from sample preparation to data reporting, are performed properly. Detection of the SPC spiked at concentrations slightly above the MT allows to determine whether pathogens are present at or above the MT. SPC was especially useful for validating the analysis of culture-negative samples that yielded only a few reads. However, we could not validate 20.7% of individual negative SOI detections, mainly because of the competition effect by reads generated when other SOI(s) were detected at high concentrations in the same sample. This should have a limited impact on the diagnosis as it would not affect the detection and identification of the major pathogen causing the infection.

We have also demonstrated that SPC can be used as a calibrator in mNGS, allowing absolute quantification of *S. aureus* GEq as efficiently as quantitative PCR (Fig. 2B). This allowed us to define a metagenomics threshold to differentiate colonization from suspected infection. Our quantitative mNGS process showed good diagnostic

capabilities (specificity: 96.6%; sensitivity: >99.9%). The "false discovery rate" of 65.5% could be due, at least in part, to a unique MT defined for all bacterial species instead of individual MT (41). However, some "false-positive" mNGS results might also reflect the presence of VBNC and antibiotic persister cells and their potential for infection recurrences (44). Additional studies, including clinical assessments, will be necessary to evaluate the diagnostic values of such "false positives" or to set specific MT to each species.

Before implementing mNGS in routine clinical diagnosis, it is imperative to address several limitations. Since metagenomics can detect any organism present in a sample, it is technically challenging to validate metagenomics assays for compliance with regulatory guidelines, rules, and standards, considering the vast array of organisms potentially detectable by NGS. In this study, we have chosen not to report the results of out-of-panel detections for which no DT has been defined and no MetaPhlAn markers have been considered. To enable the clinical use of metagenomics, *in vitro* diagnostic regulation bodies need to anticipate the modalities for reporting positive results outside the validation panel, which could have clinical value.

Furthermore, in our experimental design, which relies on SPC-based pathogen quantification, we did not use typical negative controls obtained by omitting the addition of clinical samples in the mNGS workflow. Nonetheless, it is worth considering the use of controls consisting of human cells spiked with SPC as an additional validation measure in the mNGS process.

Additionally, variations in lysis efficiencies between SPC and SOIs may lead to distorted quantification of SOIs. These differences can be empirically tested for each SPC/SOI combination and taken into account to enhance the accuracy of SOI quantification. In the present study, we employed a combination of mechanical and chemical lysis, an approach that has recently demonstrated the ability to yield reasonably accurate estimates of bacterial absolute quantifications (50).

Other limitations of mNGS are the extended sample processing time, the necessity for skilled technicians and trained bioinformaticians, as well as the associated costs for NGS system installation, maintenance, and operation, which pose challenges for most routine diagnostic laboratories in terms of affordability. In our study, the mNGS time-to-result was approximately 50 hours, mostly due to the time required for Illumina MiSeq sequencing. While this timeframe is comparable to that of standard culture-based methods, it offers the advantage of detecting a wider range of bacterial species. Our SPC-based mNGS approach, one of the pioneering applications for clinical samples, streamlined the process related to the control and quantification of extracted DNA and sequencing library, resulting in a reduction of hands-on time by 4–5 hours. The use of an SPC is also compatible with the development of integrated platforms capable of performing all steps, from sample preparation to reporting, without additional intervention. Current sequencing technologies now deliver results more rapidly than those reported in our study, making it feasible to identify pathogens within a timeframe suitable for clinical practice and thus improving patient care.

## ACKNOWLEDGMENTS

The authors would like to thank Alex van Belkum, Scientific Director of Microbiology Research at bioMérieux, for constructive comments on the manuscript.

This study was investigator-initiated and sponsored by Geneva University Hospitals. An unrestricted grant from bioMérieux (Marcy l'Étoile, France) was used to financially support this study, which was independently designed, executed, analyzed, and reported by the authors.

M.T., E.R., S.S., and G. Guigon developed the bioinformatics. M.T., V. Lazarevic, S.H., E.S.A., V. Lanet, M.G., C.M., G. Gervasi, and J.S. conceived and designed the study, and performed the analysis. M.T., S.H., E.S.A., and S.S. wrote the article and all the authors reviewed it.

## AUTHOR AFFILIATIONS

[1]bioMérieux, Grenoble, France
[2]Genomic Research Laboratory, Service of Infectious Diseases, Geneva University Hospitals, Geneva, Switzerland
[3]bioMérieux, Marcy-l'Étoile, France
[4]bioMérieux, La Balme-les-Grottes, France

## AUTHOR ORCIDs

S. Hauser  http://orcid.org/0000-0001-5076-2730

## AUTHOR CONTRIBUTIONS

S. Hauser, Conceptualization, Data curation, Formal analysis, Investigation, Methodology, Project administration, Writing – original draft | V. Lazarevic, Conceptualization, Data curation, Formal analysis, Investigation, Methodology, Validation, Writing – original draft, Writing – review and editing | M. Tournoud, Conceptualization, Data curation, Formal analysis, Investigation, Methodology, Software, Validation, Writing – review and editing | E. Ruppé, Conceptualization, Data curation, Formal analysis, Investigation, Methodology, Writing – original draft, Writing – review and editing | E. Santiago Allexant, Formal analysis, Methodology, Supervision, Validation, Writing – original draft, Writing – review and editing | G. Guigon, Project administration, Software, Supervision, Writing – review and editing | S. Schicklin, Data curation, Investigation, Software, Writing – review and editing | V. Lanet, Data curation, Formal analysis, Investigation, Methodology, Writing – review and editing | M. Girard, Formal analysis, Writing – review and editing | C. Mirande, Investigation, Supervision, Writing – review and editing | G. Gervasi, Investigation, Project administration, Supervision, Writing – review and editing | J. Schrenzel, Conceptualization, Supervision, Validation, Writing – original draft, Writing – review and editing

## DATA AVAILABILITY

The data sets supporting the conclusions of this article are available in the NCBI Sequence Read Archive (SRA) repository under study number PRJNA771365. The sequence reads were deposited after filtering out the read pairs that matched the human genome.

## ETHICS APPROVAL

BAL samples were collected by the Geneva University Hospitals (Switzerland) for diagnostic purposes. Well-characterized left-over BAL samples were then used for research activities. Because of the difficulty of obtaining consent in the short time frame required for the right execution of the DNA extraction (no sample freezing possible) and because there was no interference with the normal care of the patient (no intervention, no collection of demographic or clinical data, no additional sample), an exemption of the necessity of informed consent was obtained in accordance with Art. 34 from the Swiss Federal Law on Research on Human Beings (LRH) and Art 37-40 from the Swiss Ordinance on research on humans with the exception of clinical trials (ORH).

After receiving the samples at the Genomic Research Laboratory of the Geneva University Hospitals, the DNA was promptly extracted, and any remaining materials were completely discarded.

## ADDITIONAL FILES

The following material is available online.

## Supplemental Material

**Supplemental material (Spectrum01294-23-s0001.docx).** Additional method information with supporting figures and tables.

## Open Peer Review

**PEER REVIEW HISTORY (review-history.pdf).** An accounting of the reviewer comments and feedback.

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
