## [Reviewer comments · Microbiology Spectrum]

Microbiology Spectrum

A metagenomics method for the quantitative detection of bacterial pathogens causing hospital-associated and ventilator-associated pneumonia

Sébastien HAUSER, Vladimir Lazarevic, Maud Tournoud, Etienne Ruppé, Emmanuelle Santiago-Allexant, Ghislaine Guigon, Stéphane Schicklin, Veronique Lanet, Myriam Girard, Caroline Mirande, Gaspard Gervasi, and Jacques Schrenzel

Corresponding Author(s): Sébastien HAUSER, bioMerieux SA

Review Timeline:

Submission Date:	May 9, 2023
Editorial Decision:	July 27, 2023
Revision Received:	September 29, 2023
Accepted:	September 29, 2023

Editor: Anne Jamet

Reviewer(s): The reviewers have opted to remain anonymous.

Transaction Report:

DOI: <https://doi.org/10.1128/spectrum.01294-23>

July 27, 2023

Dr. Sébastien HAUSER
bioMerieux SA
Grenoble
France

Re: Spectrum01294-23 (A metagenomics method for the quantitative detection of pathogens causing ventilator-associated pneumonia)

Dear Dr. Sébastien HAUSER:

Thank you for submitting your manuscript to Microbiology Spectrum.

We have now received all the peer reviewers' reports and we would like to invite you to revise the paper on the basis of the points listed below.

We look forward to receiving your revision.

Kind regards,
Anne Jamet

Link Not Available

Sincerely,

Anne Jamet

Journals Department
Reviewer comments:

Reviewer #1 (Comments for the Author):

The article from Hauser et al. tried to design a new pipeline for quantification of bacterial pathogens using mNGS in respiratory samples. This article helps to standardize the metagenomics process for respiratory samples considering the technological difficulties related to the high human DNA proportions and to the difficulty to develop an efficient quantitative method. The article compared the mNGS performance using a spiking bacterium to culture and qPCR for *S. aureus*. Authors showed a sensitivity > 99.9% compared to culture and tried to define several cut-off values to standardize the process. The efforts of standardization and contamination analysis done by the authors has to be highlighted (even if not perfect yet), as this remains a major limitation for mNGS data comparison.

Nevertheless, some points need to be corrected or clarified for better understanding of the paper.

Major points:

1. The authors used a *Bacillus subtilis* strain. Could you rapidly expose how is the bacterium conserved in the BioBall® kit (bacteria or only spores?). This may impact the extraction methods (and thus the quantitative results) as spores can be more difficult to extract.
2. DNA extraction method is only focusing on bacteria (elimination of viral nucleic acids during the extraction method). I would therefore add this limitation and adapt the title of the paper to better highlight the bacterial interest of the article.
3. Considering the steps before mNGS, authors used a human DNA depletion method with saponin and DNAase before extraction of total nucleic acids. Authors show in Suppl methods that *E. coli* (or *E. cloacae*?) contaminations may be possible. I am wondering if negative controls were included for identification of potential contaminations and the eventual impact of them in the pipeline. This seems to be a critical point in the pipeline as it has to be always compared to culture, whereas we know culture has also some limits (see following point).
4. All results are compared to microbiological culture (gold standard). Culture is known to give a rapid answer but subjected to inter-observer variability and intrinsic limitations (mainly detection of easy growing bacteria, no detection of anaerobes in respiratory samples...). I would, therefore, discuss this point as authors are also pointing out the bad correlation between CFU/ml and mNGS quantification but also the necessity to compare the mNGS results to culture... The good sensitivity result shown here needs also to be lowered as culture performance is easily outdated compared to metagenomics.
5. Authors tried to evaluate the EffSOI in the Additional methods, nevertheless, this coefficient should appear and be explained in the main text because it represents a true limit of the calculation depending on the extraction method and because it has not been evaluated for all SOI. This has also to be discussed.
6. For the 20 samples with negative results (such as sample 4, 5, 8...), did authors find other potential bacteria that were not part of the 20 pathogens? I know a 16S/MetaPhlan2 analysis was used, but this should be better explained in the text as 50% of the samples of the validation set were negative. Do you have any idea if this was correlated with clinical data and if pneumonia was retrospectively confirmed?

Minor points:

1. For a better comprehension of the article that uses a lot of abbreviation and ratios, I would recommend to replace Figure 1 by Additional Figure 2.
2. Line 287: Sensitivity value should be changed to "> 99.9%".
3. Table 2: why is culture quantification of sample T01 going up to 6 whereas all other sample are > 5?
4. Figure 1: time from sample to result could be shown, as this can still be a major concern for routine application of such pipeline.
5. Why only focusing on *S. aureus* qPCR correlation? Could we expect the same type of result for *H. influenzae* or *Enterobacterales*?

Reviewer #2 (Comments for the Author):

The article by Hauser et al. describes a metagenomics NGS (mNGS) approach for the quantitative detection of 20 relevant bacteria from bronchoalveolar samples, in the context of suspicion of pneumoniae, with culture used as a gold standard for performance evaluation. mNGS is increasingly used for routine microbiological diagnosis but still faces issues of method

standardization and quantification challenges, what makes this work deal with a subject of interest for many readers. The paper is well-written, and the methodology demonstrates a certain conceptual depth. My remarks do not really relate to the scientific or methodological quality of the work, but rather to its philosophy and its purpose. This is the reason why I would appreciate if the authors could respond to some or all of them before publication.

Main remarks:

- What is the point of doing mNGS, which is costly, quite long and requires significant expertise, when the goal is the detection/diagnosis of just 20 clinically relevant bacteria causing pneumoniae? Why not qPCR instead? The BioFire FilmArray respiratory panel can detect 22 targets in less than 1h starting from biological sample.
- Maybe this study must be seen as a first stage of development before a broad-range pathogens NGS test comes. In that case, to which extent can the quantification method presented here be generalized to many more bacteria, then to DNA and RNA viruses, or even to other type of pathogens causing pneumoniae?
- As far as I understand the normalization procedure (sorry if I am wrong): the absolute quantification method is valid assuming that the sum of all SOI reads + the sum of SPC = the sum of all bacterial reads within the dataset. Indeed, line 173, the content of the "internal reference database" is not very clear to me. What would be the consequence, in term of quantification, of a co-infection by one of the 20 SOI + one extra bacteria which is not included in the 20 SOI list nor in the "internal database"? I think that in such a case, the quantification would not work. Can you please comment on that?
- The Positive Predictive Value (PPV) is very low (34.5%), questioning the clinical relevance of the test. We can anticipate the PPV to be even lower as the scope of the pathogens of interest increases. What do you think about it and what are the perspectives?

Minor remarks:

- What were the criteria for establishing the list of 20 bacteria? Why is there no overlap with the Biofire respiratory panel list?
- Please indicate the turnaround time of your mNGS approach, from sample to result. Miseq sequencing in 2x250 is >24h, not including upstream library prep. So probably not competitive compared to culture, and certainly not competitive compared to fast qPCR.
- Bioinformatics tools seem outdated: kraken (now kraken2), idba assembler (2012), metaphlan2 (now metaphlan4).
- Line 118: For non-expert readers, can you describe briefly which genera or species can be detected with the culture media mentioned? In particular, are all the 20 SOI detectable by such culture media?
- Line 169: ref 29 Tournoud et al. (bioRxiv), please state in the text that this reference is a preprint waiting for peer-review, or remove it.
- Line 184: "genome coverage depth" is not very clear to me. Please clarify, either vertical coverage, or horizontal coverage.
- Line 263: Would make sense to add T07 to Figure 2B.
- Figure 2: Please add the full equation of the linear regression and comment on the slope.
- Line 363: Please add references for VBNC.
- Authors self-citations: Ruppé cited 4 times, potentially abusive.

Staff Comments:

Preparing Revision Guidelines

- Point-by-point responses to the issues raised by the reviewers in a file named "Response to Reviewers," NOT IN YOUR

COVER LETTER.

- Upload a compare copy of the manuscript (without figures) as a "Marked-Up Manuscript" file.
- Each figure must be uploaded as a separate file, and any multipanel figures must be assembled into one file.
- Manuscript: A .DOC version of the revised manuscript
- Figures: Editable, high-resolution, individual figure files are required at revision, TIFF or EPS files are preferred

Please return the manuscript within 60 days; if you cannot complete the modification within this time period, please contact me. If you do not wish to modify the manuscript and prefer to submit it to another journal, please notify me of your decision immediately so that the manuscript may be formally withdrawn from consideration by Microbiology Spectrum.

The article from Hauser et al. tried to design a new pipeline for quantification of bacterial pathogens using mNGS in respiratory samples. This article helps to standardize the metagenomics process for respiratory samples considering the technological difficulties related to the high human DNA proportions and to the difficulty to develop an efficient quantitative method.

The article compared the mNGS performance using a spiking bacterium to culture and qPCR for *S. aureus*. Authors showed a sensitivity > 99.9% compared to culture and tried to defined several cut-off values to standardize the process. The efforts of standardization and contamination analysis done by the authors has to be highlight (even if not perfect yet), as this remains a major limitation for mNGS data comparison.

Nevertheless, some points need to be corrected or clarified for better understanding of the paper.

Major points:

1. The authors used a *Bacillus subtilis* strain. Could you rapidly expose how is the bacterium conserved in the BioBall® kit (bacteria or only spores?). This may impact the extraction methods (and thus the quantitative results) as spores can be more difficult to extract.
2. DNA extraction method is only focusing on bacteria (elimination of viral nucleid acids during the extraction method). I would therefore add this limitation and adapt the title of the paper to better highlight the bacterial interest of the article.
3. Considering the steps before mNGS, authors used a human DNA depletion method with saponin and DNAase before extraction of total nucleic acids. Authors show in Suppl methods that *E. coli* (or *E. cloacae*?) contaminations may be possible. I am wondering if negative controls were included for identification of potential contaminations and the eventual impact of them in the pipeline. This seems to be a critical point in the pipeline as it has to be always compared to culture, whereas we know culture has also some limits (see following point).
4. All results are compared to microbiological culture (gold standard). Culture is known to give a rapid answer but subjected to inter-observer variability and intrinsic limitations (mainly detection of easy growing bacteria, no detection of anaerobes in respiratory samples...). I would, therefore, discuss this point as authors are also pointing out the bad correlation between CFU/ml and mNGS quantification but also the necessity to compare the mNGS results to culture... The good sensitivity result shown here needs also to be lowered as culture performance is easily outdated compared to metagenomics.
5. Authors tried to evaluate the EffSOI in the Additional methods, nevertheless, this coefficient should appear and be explained in the main text because it represents a true limit of the calculation depending on the extraction method and because it has not been evaluated for all SOI. This has also to be discussed.

6. For the 20 samples with negative results (such as sample 4, 5, 8...), did authors find other potential bacteria that were not part of the 20 pathogens? I know a 16S/MetaPhlan2 analysis was used, but this should be better explained in the text as 50% of the samples of the validation set were negative. Do you have any idea if this was correlated with clinical data and if pneumonia was retrospectively confirmed?

Minor points:

1. For a better comprehension of the article that uses a lot of abbreviation and ratios, I would recommend to replace Figure 1 by Additional Figure 2.

2. Line 287: Sensitivity value should be changed to "> 99.9%".

3. Table 2: why is culture quantification of sample T01 going up to 6 whereas all other sample are > 5?

4. Figure 1: time from sample to result could be shown, as this can still be a major concern for routine application of such pipeline.

5. Why only focusing on *S. aureus* qPCR correlation? Could we expect the same type of result for *H. influenzae* or *Enterobacterales*?

The article by Hauser et al. describes a metagenomics NGS (mNGS) approach for the quantitative detection of 20 relevant bacteria from bronchoalveolar samples, in the context of suspicion of pneumoniae, with culture used as a gold standard for performance evaluation. mNGS is increasingly used for routine microbiological diagnosis but still faces issues of method standardization and quantification challenges, what makes this work deal with a subject of interest for many readers. The paper is well-written, and the methodology demonstrates a certain conceptual depth. My remarks do not really relate to the scientific or methodological quality of the work, but rather to its philosophy and its purpose. This is the reason why I would appreciate if the authors could respond to some or all of them before publication.

Main remarks:

- What is the point of doing mNGS, which is costly, quite long and requires significant expertise, when the goal is the detection/diagnosis of just 20 clinically relevant bacteria causing pneumoniae? Why not qPCR instead? The BioFire FilmArray respiratory panel can detect 22 targets in less than 1h starting from biological sample.
- Maybe this study must be seen as a first stage of development before a broad-range pathogens NGS test comes. In that case, to which extent can the quantification method presented here be generalized to many more bacteria, then to DNA and RNA viruses, or even to other type of pathogens causing pneumoniae?
- As far as I understand the normalization procedure (sorry if I am wrong): the absolute quantification method is valid assuming that the sum of all SOI reads + the sum of SPC = the sum of all bacterial reads within the dataset. Indeed, line 173, the content of the "internal reference database" is not very clear to me. What would be the consequence, in term of quantification, of a co-infection by one of the 20 SOI + one extra bacteria which is not included in the 20 SOI list nor in the "internal database"? I think that in such a case, the quantification would not work. Can you please comment on that?
- The Positive Predictive Value (PPV) is very low (34.5%), questioning the clinical relevance of the test. We can anticipate the PPV to be even lower as the scope of the pathogens of interest increases. What do you think about it and what are the perspectives?

Minor remarks:

- What were the criteria for establishing the list of 20 bacteria? Why is there no overlap with the Biofire respiratory panel list?
- Please indicate the turnaround time of your mNGS approach, from sample to result. Miseq sequencing in 2x250 is >24h, not including upstream library prep. So probably not competitive compared to culture, and certainly not competitive compared to fast qPCR.
- Bioinformatics tools seem outdated: kraken (now kraken2), idba assembler (2012), metaphlan2 (now metaphlan4).
- Line 118: For non-expert readers, can you describe briefly which genera or species can be detected with the culture media mentioned? In particular, are all the 20 SOI detectable by such culture media?

- Line 169: ref 29 Tournoud et al. (bioRxiv), please state in the text that this reference is a preprint waiting for peer-review, or remove it.
- Line 184: “genome coverage depth” is not very clear to me. Please clarify, either vertical coverage, or horizontal coverage.
- Line 263: Would make sense to add T07 to Figure 2B.
- Figure 2: Please add the full equation of the linear regression and comment on the slope.
- Line 363: Please add references for VBNC.
- Authors self-citations: Ruppé cited 4 times, potentially abusive.

Point-by-point responses to reviewers.

Dear reviewers, thank you for your interest in our manuscript. You will find below all your comments followed, in blue, by our answers.

Reviewer #1 (Comments for the Author):

The article from Hauser et al. tried to design a new pipeline for quantification of bacterial pathogens using mNGS in respiratory samples. This article helps to standardize the metagenomics process for respiratory samples considering the technological difficulties related to the high human DNA proportions and to the difficulty to develop an efficient quantitative method.

The article compared the mNGS performance using a spiking bacterium to culture and qPCR for *S. aureus*. Authors showed a sensitivity > 99.9% compared to culture and tried to define several cut-off values to standardize the process. The efforts of standardization and contamination analysis done by the authors has to be highlight (even if not perfect yet), as this remains a major limitation for mNGS data comparison.

Nevertheless, some points need to be corrected or clarified for better understanding of the paper.

Major points:

1. The authors used a *Bacillus subtilis* strain. Could you rapidly expose how is the bacterium conserved in the BioBall® kit (bacteria or only spores?). This may impact the extraction methods (and thus the quantitative results) as spores can be more difficult to extract.

Response: We appreciate the reviewer's input on this matter. We acknowledge that spores may be more difficult to disrupt than some vegetative cells. However, we employed a rigorous mechanical lysis method, utilizing both 1 mm and 0.1 mm beads, combined with chemical lysis, for an effective disruption.

*It is worth noting that the commercial Cepheid GeneXpert Dx System uses dried *Bacillus globigii* spores as the sample processing control (SPC). Of course, it is important to consider that the cutoffs of clinical significance for pathogen amounts might need to be adjusted based on the specific SPC species and types, whether they are vegetative cells, spores, lyophilized, frozen, or fresh. In the Additional file, we introduced the concept of Eff_{SOI} , which accounts for differences in lysis efficiency between the SPC and SOIs. This factor can be empirically determined for each SPC/SOI combination based on a large number of samples.*

2. DNA extraction method is only focusing on bacteria (elimination of viral nucleid acids during the extraction method). I would therefore add this limitation and adapt the title of the paper to better highlight the bacterial interest of the article.

Response: We appreciate the reviewer for their feedback on this important point. We have now updated the title to emphasize the primary focus of the article on bacteria. We also pointed it out in the Background section.

3. Considering the steps before mNGS, authors used a human DNA depletion method with saponin and DNAase before extraction of total nucleic acids. Authors show in Suppl methods that E. coli (or E. cloacae?) contaminations may be possible. I am wondering if negative controls were included for identification of potential contaminations and the eventual impact of them in the pipeline. This seems to be a critical point in the pipeline as it has to be always compared to culture, whereas we know culture has also some limits (see following point).

Response: We had considered using a negative control by omitting the addition of the clinical sample in the mNGS workflow. However, the DNA yield in the negative control extracts is significantly lower than the requirements of the sequencing library preparation kits. Under these conditions, the sequencing of a negative control is inefficient.

We made a different technical choice by devising a detection threshold (DT) for each species, as explained in the Additional file. For each of the species of interest (SOI), we sequenced DNA from BAL samples that tested negative in culture for that specific SOI, measured the background of sequences classified as SOI and applied the 3-sigma method to obtain the DTSOI. However, recognizing the need for quantitative data for VAP diagnosis, we employed, instead, SPC as a calibrator for quantitative detection. We only reported SOIs with concentrations above the metagenomic threshold (MT) as positive.

Regarding E. cloacae, it appears to be a contamination at the in silico level, revealing the importance of utilizing high-quality, validated reference databases. As for E. coli, we have indeed mentioned a possibility of reagent contamination. However, in the samples in question, the abundance of identified E. coli reads suggests the true presence of E. coli or the potential misclassification of a fraction of reads from Klebsiella as E. coli.

Therefore, in response to the reviewer's comment, we have now added a statement in the discussion section to address the issue of negative controls.

4. All results are compared to microbiological culture (gold standard). Culture is known to give a rapid answer but subjected to inter-observer variability and intrinsic limitations (mainly detection of easy growing bacteria, no detection of anaerobes in respiratory samples...). I would, therefore, discuss this

point as authors are also pointing out the bad correlation between CFU/ml and mNGS quantification but also the necessity to compare the mNGS results to culture... The good sensitivity result shown here needs also to be lowered as culture performance is easily outdated compared to metagenomics.

Response: We received the quantification results by culture at the nearest log₁₀, which does not allow straightforward comparisons with concentrations measured by molecular tests. For samples with high bacterial loads, the results received from the bacteriology laboratory were reported as '>10⁵'. However, this lack of precision in the microbiological culture reports is not the only reason for the lack of correlation with mNGS quantification. This discrepancy is also attributable to the fact that culture detects live and growing bacteria (CFU = colony forming unit), while molecular methods such as NGS and PCR also detect DNA (GEq = genome equivalent) of bacteria that do not grow (whatever the reason). In addition, some bacteria tend to form clusters that will produce a single colony while each cell from the cluster will have a complete genome detectable by molecular techniques. It is therefore not possible to establish a direct correlation between CFU and GEq. We have now incorporated these aspects in the discussion section.

The comparison to culture allows to check that mNGS is, at least, able to detect all the pathogens identified by culture, which we achieved with a sensitivity >99.9%. Additional metagenomic detections have been classified as "false positives", considering culture as the reference method. The clinical value of these "false positives" remains to be established but appears to be of potential significance.

5. Authors tried to evaluate the EffSOI in the Additional methods, nevertheless, this coefficient should appear and be explained in the main text because it represents a true limit of the calculation depending on the extraction method and because it has not been evaluated for all SOI. This has also to be discussed.

Response: We did not use the EffSOI factor for the calculations presented in this article. This factor would make it possible to gain in precision by considering the difference in extraction and sequencing efficiencies between the calibrator and the quantified species. The quantity of sample positives for the species studied was insufficient to allow the evaluation of EffSOI.

In accordance with the reviewer's suggestion, we have added a section on the differential lysis efficiency to the discussion. It is important to note that the inclusion of a reference pertaining to this additional content results in an increase in self-citations.

6. For the 20 samples with negative results (such as sample 4, 5, 8...), did authors find other potential bacteria that were not part of the 20 pathogens? I know a 16S/MetaPhlan2 analysis was used, but this should be better explained in the text as 50% of the samples of the validation set were negative. Do you have any idea if this was correlated with clinical data and if pneumonia was retrospectively confirmed?

Response: Unfortunately, for the purpose of this study, we only had access to leftover BAL samples and microbial culture results, but not to the patient's clinical documentation. We did detect microbes other than the 20 SOIs but not only exclusively in negative samples. These represented species of the 'normal' oral/respiratory flora. However, we focused only on the pathogen panel, so 16S/ MetaPhlan markers of other ('normal' flora) species were not considered.

Minor points:

1. For a better comprehension of the article that uses a lot of abbreviation and ratios, I would recommend to replace Figure 1 by Additional Figure 2.

Response: In response to the reviewer's comment, we propose to merge the two figures for easier comprehension of the manuscript.

2. Line 287: Sensitivity value should be changed to "> 99.9%".

Response: We have now made the requested modification in the document.

3. Table 2: why is culture quantification of sample T01 going up to 6 whereas all other sample are > 5?

Response: We appreciate the reviewer's observation and thank them for pointing out the discrepancy in Table 2 regarding the culture quantification of sample T01. We acknowledge that the data values as originally provided appear non-uniform. To ensure clarity in our presentation, we have made the necessary adjustment by changing the value '6' to '>5' in Table 2. Additionally, corresponding changes have been made in Figure 2A and in the main text to maintain consistency throughout the manuscript.

4. Figure 1: time from sample to result could be shown, as this can still be a major concern for routine application of such pipeline.

Response: "In response to the reviewer's suggestion, we have incorporated the time from sample to result in Figure 1 and have addressed this aspect in the discussion section.

5. Why only focusing on *S. aureus* qPCR correlation? Could we expect the same type of result for *H. influenzae* or *Enterobacterales*?

*Response: We acknowledge that expanding our analysis to include other species would provide a more comprehensive understanding of the mNGS-qPCR correlation. However, the limitations of sample availability restricted our ability to explore correlations beyond *S. aureus*. We lacked a sufficient number of positive samples for other species to conduct relevant analyses.*

Reviewer #2 (Comments for the Author):

The article by Hauser et al. describes a metagenomics NGS (mNGS) approach for the quantitative detection of 20 relevant bacteria from bronchoalveolar samples, in the context of suspicion of pneumoniae, with culture used as a gold standard for performance evaluation. mNGS is increasingly used for routine microbiological diagnosis but still faces issues of method standardization and quantification challenges, what makes this work deal with a subject of interest for many readers. The paper is well-written, and the methodology demonstrates a certain conceptual depth. My remarks do not really relate to the scientific or methodological quality of the work, but rather to its philosophy and its purpose. This is the reason why I would appreciate if the authors could respond to some or all of them before publication.

Main remarks:

1. What is the point of doing mNGS, which is costly, quite long and requires significant expertise, when the goal is the detection/diagnosis of just 20 clinically relevant bacteria causing pneumoniae? Why not qPCR instead? The BioFire FilmArray respiratory panel can detect 22 targets in less than 1h starting from biological sample.

This R&D project aims to demonstrate the use of metagenomics for clinical diagnostics with

quantitative detection and sample processing control. As it stands, it is still premature to use this workflow for a routine clinical test. Since the completion of this work, we have been able to continue to progress at the same time as sequencing techniques evolve to reduce considerably both the duration of sequencing and its cost. We are confident that this dynamic reduction of time to results and cost will continue making sequencing assays competitive for routine use in a near future.

Metagenomics can detect any organism present in a sample; however, it is not technically possible to validate metagenomics assays on all potentially detectable organisms by sequencing. This is why we reduced our panel of organisms to the 20 most frequently encountered species. Today there is no clear guidelines from IVD regulation bodies on how clinical metagenomic tests should be validated and how to report a result for pathogens on which the tests have not been validated.

From now, we can find an interest in the use of a metagenomic test, especially in the case of atypical infections. While it is true that BioFire FilmArray is able to detect 22 different pathogens, when the test is negative no information on the presence of a pathogen out from the respiratory panel can be provided. In addition, metagenomics has the ability to reconstruct the genome of the detected pathogen to provide more information such as antibiotic resistance, virulence, phylogeny ...

2. Maybe this study must be seen as a first stage of development before a broad-range pathogens NGS test comes. In that case, to which extend can the quantification method presented here be generalized to many more bacteria, then to DNA and RNA viruses, or even to other type of pathogens causing pneumoniae?

Today, IVD regulation bodies remain unclear on how agnostic tests such as metagenomics should be validated. The sequence database used in this study contains more than 10,000 genomes. Our metagenomic tests therefore can detect several thousand different species, so it is not possible to validate the detection and quantification of all potentially detectable species with the published test. The idea is therefore to make a minimum validation on the most frequent species encountered during the VAP. This test also does not claim to precisely quantify the concentration of pathogen in the sample. The objective is to have a quantification at the nearest Log level for comparison to a clinical decision threshold.

3. As far as I understand the normalization procedure (sorry if I am wrong): the absolute quantification method is valid assuming that the sum of all SOI reads + the sum of SPC = the sum of all bacterial reads within the dataset. Indeed, line 173, the content of the "internal reference database" is not very clear to me. What would be the consequence, in term of quantification, of a co-infection by one of the 20 SOI + one extra bacteria which is not included in the 20 SOI list nor in the "internal database"? I think that in such a case, the quantification would not work. Can you please comment on that?

Response: The sequence database used for taxonomic classification of sequence reads was not limited to genomes of the 20 SOIs, it also included genomes from the organisms found in lung and oral cavity. Thus, the sum of reads associated to SOI or SPC was not equal to the sum of reads associated

to all bacteria. We chose to normalize the quantities of reads in relation to the amount of all bacterial reads (SOI, SPC and other species from the database), because between the samples there is a large variability in the amount of DNA sequenced, in the number of reads produced by sequencing run and on the ratio in quantity between the patient's DNA and the microbial DNA. We chose not to report the results of non-SOI detections because they were not validated with defined thresholds and MetaPhlAn markers.

4. The Positive Predictive Value (PPV) is very low (34.5%), questioning the clinical relevance of the test. We can anticipate the PPV to be even lower as the scope of the pathogens of interest increases. What do you think about it and what are the perspectives?

Response: Yes, the PPV is low because we have a 'false positive' rate that is high. As answered in the "major comment" number 4 of the first reviewer, this is due to the fact that we compared culture results with molecular results and that there is no direct correlation between the number of bacterial cells capable of growing on the media and the number of bacterial genomes quantity. We expected to detect more pathogens by mNGS than by culture, especially in patients treated with antibiotics. Integration of additional clinical data is needed to enhance the accuracy of our test aimed at improving the clinical relevance of our test, with the goal of reducing false positives and increasing its utility in clinical practice.

Minor remarks:

1. What were the criteria for establishing the list of 20 bacteria? Why is there no overlap with the Biofire respiratory panel list?

Response: The BioFire FILMARRAY Pneumonia plus Panel detects and identifies most common pathogens commonly associated with respiratory infections, including viruses. The list of bacteria in our study, established in collaboration with our colleagues at the Geneva University Hospitals (HUG), comprises the 20 most frequently detected species in cases of a specific respiratory infections, i.e., ventilator-acquired pneumonia (VAP). Fourteen of these species overlap with those covered by the BioFire FILMARRAY Pneumonia plus Panel.

2. Please indicate the turnaround time of your mNGS approach, from sample to result. Miseq sequencing in 2x250 is >24h, not including upstream library prep. So probably not competitive compared to culture, and certainly not competitive compared to fast qPCR.

Response: The need to reduce the turnaround time is already discussed at the end of the conclusion. A new version integrating the duration of sequencing steps is proposed in the answer of the first minor point from the first reviewer.

3. Bioinformatics tools seem outdated: kraken (now kraken2), idba assembler (2012), metaphlan2 (now metaphlan4).

Response: Although new tool versions have been released since our work have been done, using them would not drastically change or improve our results. kraken2 would allow us to reduce the calculation time while it will return the exact same results, moving toward another assembler like Spades or Skesa would give close assemblies, and we believe the metaphlan updates are mainly focused on the understanding of the gut microbiome diversity, it may help a bit, but we guess it will not be significant.

4. Line 118: For non-expert readers, can you describe briefly which genera or species can be detected with the culture media mentioned? In particular, are all the 20 SOI detectable by such culture media?

Response: The 20 species of interest (SOI) can grow on the media used for culturing the respiratory specimens except for Legionella pneumophila. Exceptionally, L. pneumophila can be grown on chocolate agar media, but after a long incubation time, and perhaps when a specimen rich in nutrients is plated (e.g. a biopsy) [ref <https://www.ncbi.nlm.nih.gov/pmc/articles/PMC4609723/>]. Here, agar media were incubated for 48h, hence L. pneumophila would not be isolated, even if nutrients were putatively brought in the specimen.

5. Line 169: ref 29 Tournoud et al. (bioRxiv), please state in the text that this reference is a preprint waiting for peer-review, or remove it.

Response: We have now modified the text to indicate that reference is a preprint awaiting peer-review

6. Line 184: "genome coverage depth" is not very clear to me. Please clarify, either vertical coverage, or horizontal coverage.

Response: We have now clarified the issue by changing the text to read "average genome sequencing depth" (thus vertical).

7. Line 263: Would make sense to add T07 to Figure 2B.

Response: Please see next response

8. Figure 2: Please add the full equation of the linear regression and comment on the slope.

Response: The comparison nearest Log of quantities concentrations determined by mNGS (in GEq/mL) and by microbial culture (in CFU/mL) (Figure 2A) showed moderate correlation (slope =

0.6087) with weak linear association poor ($R^2 = 0.3043$) correlation that are poorly statistically significant ($p = 0.156$) (Figure 2A). The comparison of the Log values of concentrations determined by mNGS and quantitative PCR (Figure 2B) showed strong correlation (slope = 1) with strong linear associations between values ($R^2 = 0.8392$) that are statistically significant ($p < 0.01$).

We observed a better correlation ($R^2 = 0.9904$) when we compared quantification results of mNGS and qPCR (Figure 2B). Moreover, for sample T07 that was culture-negative, both qPCR and mNGS detected and quantified *S. aureus* above 10^4 GEq/mL.

New figure 2 version has now been provided.

Figure 2: Scatter plot of *Staphylococcus aureus* quantification by (A) Microbial culture in CFU/mL or (B) quantitative PCR in GEq/mL vs. metagenomics sequencing in GEq/mL. In A, the plotted values are the nearest Log of concentrations as provided by microbiologists for culture quantification. The diamond corresponds to the mNGS quantification of *S. aureus* in a culture negative sample (sample T07). The squares correspond to mNGS quantification of sample for which the concentration of *S. aureus* provided by microbiologists were reported as "> 5". pairs of overlapping plot symbols are indicated by "(2)". n indicates the number of observations. In B, the plotted values are the Log of the calculated concentrations.

9. Line 363: Please add references for VBNC.

Response: References 42 and 43 have now been moved closer to the VBNC acronym.

10 Authors self-citations: Ruppé cited 4 times, potentially abusive.

Response: Dr. Ruppé has played an active and instrumental role in the development and implementation of metagenomics in clinical practice. Therefore, we believe that citing his manuscripts (on only two of which he stands as a first author) is not only logical but also relevant to the research presented. Furthermore, it is important to note that among the four (five in the revised version) cited papers, one is a preprint that outlines the methodology employed in our current study. This preprint

serves as a foundational reference for our research and provides essential context for some methodological aspects of the present manuscript. One of the cited papers is directly related to the wet lab work, specifically regarding mechanical cell disruption. This inclusion was made in response to questions raised by Reviewer #1 during the manuscript revision process. Additionally, two of the cited papers are highly pertinent to the present topic, as they deal with the role of metagenomics in pneumonia. One paper places our research within a broader framework, encompassing aspects such as culprits, expectations, and data sharing in clinical metagenomics. Therefore, we believe that the inclusion of these citations serve to strengthen the foundation and validity of our findings, so we prefer that they are retained in the manuscript.

September 29, 2023

Dr. Sébastien HAUSER
bioMerieux SA
Grenoble
France

Re: Spectrum01294-23R1 (A metagenomics method for the quantitative detection of bacterial pathogens causing hospital-associated and ventilator-associated pneumonia)

Dear Dr. Sébastien HAUSER:

Your manuscript has been accepted, and I am forwarding it to the ASM Journals Department for publication. You will be notified when your proofs are ready to be viewed.

Sincerely,

Anne Jamet
Editor, Microbiology Spectrum
